# Energy-saving fast-forward scaling

**Takuya Hatomura**

NTT Basic Research Laboratories & NTT Research Center for
Theoretical Quantum Information, NTT Corporation, Kanagawa 243-0198, Japan

takuya.hatomura@ntt.com

## Abstract

We propose energy-saving fast-forward scaling. Fast-forward scaling is a method which enables us to speed up (or slow down) given dynamics in a certain measurement basis. We introduce energy costs of fast-forward scaling, and find possibility of energy-saving speedup for time-independent measurement bases. As concrete examples, we show such energy-saving fast-forward scaling in a two-level system and quantum annealing of a general Ising spin glass. We also discuss the influence of a time-dependent measurement basis, and give a remedy for unwanted energy costs. The present results pave the way for realization of energy-efficient quantum technologies.

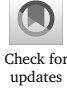
# 1 Introduction

Quantum technologies have the ability to outperform classical technologies [1]. Recently, various quantum devices were manufactured and basic quantum information processing was demonstrated [2–4]. However, implementation of practical quantum algorithms [5, 6] on given quantum devices is still challenging. It requires fast control of dynamics because decoherence causes quantum-to-classical transitions [7].

Shortcuts to adiabaticity were proposed as promising means of fast control, which enable us to speed up intrinsically slow adiabatic time evolution [8–11]. There are various approaches. In counterdiabatic driving, fast-forwarding of adiabatic time evolution is realized by applying control fields and/or interactions which cancel out nonadiabatic transitions [12–14]. In invariant-based inverse engineering, we tailor a dynamical mode by scheduling a given Hamiltonian so that the initial state and the final state coincide with these of adiabatic time evolution [15]. In fast-forward scaling, we realize fast-forwarding of any dynamics in a certain measurement basis by inverse engineering of a Hamiltonian [16]. Notably, it can also be applied to speedup of adiabatic time evolution [17].

There exists tradeoff relation between speed of dynamics and its energy [18]. Accordingly, realization of high-speed control requires a high energy cost. It is a fair question to ask the energy costs of shortcuts to adiabaticity. There are limitations on experimentally realizable energy scale, and thus evaluation of energy costs is necessary to estimate the maximum speed of dynamics via shortcuts to adiabaticity. It is also of great importance to discuss how efficiently we can realize speedup of target dynamics. Various figures of merit for energy costs of shortcuts to adiabaticity have been proposed and analyzed [19–23].

In this paper, we discuss energy costs of fast-forward scaling. The Hilbert-Schmidt norm of additional terms is often used as an energy cost of quantum control [20, 21]. However, in fast-forward scaling, an original Hamiltonian is also modulated as well as additional driving is introduced. The Hilbert-Schmidt norm of a total Hamiltonian, which was introduced to evaluate energy costs of counterdiabatic driving [19, 23], is more suitable for an energy cost of fast-forward scaling. We introduce an instantaneous energy cost and a total energy cost by using this quantity. Then, we consider reducing these energy costs. We find that these energy costs can easily be minimized when a measurement basis does not depend on time. We also study the influence of a time-dependent measurement basis and give a remedy for unwanted energy costs. Our main finding is the existence of nontrivial speedup of target dynamics with reduced energy costs, which we propose as *energy-saving fast-forward scaling*.

This paper is constructed as follows. First, we introduce the simplest fast-forward scaling and general fast-forward scaling in Sec. 2. Next, we introduce the energy costs of fast-forward scaling in Sec. 3. We define the energy costs of the simplest fast-forward scaling as the standard energy costs. Then, in Sec. 4, we show that there exist nontrivial speeding-up protocols which require lower energy costs than the standard energy costs. In Sec. 5, we apply energy-saving fast-forward scaling to a two-level system and quantum annealing in a general Ising spin glass. We also discuss influence of a time-dependent measurement basis on fast-forward scaling of a two-level system. We give a way of reducing unwanted energy costs and insights into general cases. We discuss the present results in Sec. 6 and summarize the results in Sec. 7.

# 2 Fast-forward scaling

First, we introduce fast-forward scaling. Here, we adopt formulation introduced in Ref. [24, 25]. Suppose that dynamics $|\Psi(t)\rangle$ is governed by a time-dependent Hamiltonian $\hat{H}(t)$ and the final state $|\Psi(T)\rangle$ at the operation time $t = T$ is a target state. We discuss state preparation of this target state within a shorter time $T_{\mathrm{FF}}$, which we call the fast-forward time.

The simplest way of fast-forward scaling is the introduction of rescaled time $s = s(t)$, where $s(0) = 0$ and $s(T_{\text{FF}}) = T$, and amplification of the Hamiltonian with the rescaled time

$$\hat{H}_{\text{FF}}(t) = \frac{ds}{dt}\hat{H}(s). \tag{1}$$

At each time, the speed of dynamics is $(ds/dt)$-times faster than that of the original dynamics, while it requires a $(ds/dt)$-times higher energy cost as discussed later. This is the simplest example of fast-forward scaling.

More generally, fast-forward dynamics is described by the Schrödinger equation

$$i\hbar \frac{\partial}{\partial t}|\Psi_{\text{FF}}(t)\rangle = \hat{H}_{\text{FF}}(t)|\Psi_{\text{FF}}(t)\rangle, \tag{2}$$

where the fast-forward state $|\Psi_{\text{FF}}(t)\rangle$ is given by

$$|\Psi_{\text{FF}}(t)\rangle = \hat{U}_f(t)|\Psi(s)\rangle, \tag{3}$$

and the fast-forward Hamiltonian $\hat{H}_{\text{FF}}(t)$ is given by

$$\hat{H}_{\text{FF}}(t) = \frac{ds}{dt}\hat{U}_f(t)\hat{H}(s)\hat{U}_f^\dagger(t) + i\hbar(\partial_t \hat{U}_f(t))\hat{U}_f^\dagger(t). \tag{4}$$

For a given complete orthonormal measurement basis $\{|\sigma\rangle\}$, we can obtain the same measurement outcome within a different time

$$|\langle\sigma|\Psi_{\text{FF}}(t)\rangle|^2 = |\langle\sigma|\Psi(s)\rangle|^2, \tag{5}$$

when the unitary operator $\hat{U}_f(t)$ is given by

$$\hat{U}_f(t) = e^{i\sum_\sigma f_\sigma(t)|\sigma\rangle\langle\sigma|}, \tag{6}$$

with an arbitrary real function $f_\sigma(t)$. Note that the measurement basis $\{|\sigma\rangle\}$ may depend on time. The above simplest example corresponds with the case where the unitary operator is given by the identity operator $\hat{U}_f(t) = \hat{1}$.

Note that we can also realize slowing-down, pausing, and rewinding of target dynamics by setting the magnification factor $ds/dt$ as $0 < ds/dt < 1$, $ds/dt = 0$, and $ds/dt < 0$, respectively. In this paper, we focus on fast-forwarding of target dynamics because a cost of speedup is of interest.

## 3 Energy costs of fast-forward scaling

Next, we introduce energy costs of fast-forward scaling. In this paper, we use the Hilbert-Schmidt norm of Hamiltonians as a component of energy costs, where the Hilbert-Schmidt norm of an Hermitian operator $\hat{H}$ is given by

$$\|\hat{H}\| = \sqrt{\text{Tr}\,\hat{H}^2}. \tag{7}$$

We define an instantaneous energy cost of fast-forward scaling as

$$\delta C(t) = \frac{\|\hat{H}_{\text{FF}}(t)\|}{\|\hat{H}(s)\|}, \tag{8}$$

and a total energy cost of fast-forward scaling as

$$C = \frac{\int_0^{T_{\text{FF}}} dt\, \|\hat{H}_{\text{FF}}(t)\|}{\int_0^{T} dt\, \|\hat{H}(t)\|}.$$

(9)

The former quantity is the instantaneous energy cost for $(ds/dt)$-times speedup, which can be used to determine the maximum speed of control with limited energy scale, and the latter quantity is the total energy cost compared with that of the original process, which can be used to evaluate energy efficiency of control.

We adopt the energy costs of the simplest fast-forward Hamiltonian (1) as the standard energy costs of fast-forward scaling. The standard instantaneous energy cost is given by

$$\delta C_{\$}(t) = \left| \frac{ds}{dt} \right|.$$

(10)

The standard total energy cost is given by

$$C_{\$} = 1,$$

(11)

when $(ds/dt) \geq 0$ for all time $t$. As mentioned before, the former quantity represents the $(ds/dt)$-times higher energy cost for $(ds/dt)$-times faster dynamics. The latter quantity implies that the simplest fast-forward scaling does not require the additional total energy cost as long as rewinding of target dynamics is not considered. Later, we compare the energy costs of general fast-forward scaling with these standard energy costs, and thus we use the symbol \$ to emphasize these standard quantities.

## 4 Energy-saving fast-forward scaling

Then, we discuss the energy costs of general fast-forward scaling. We find that the squared Hilbert-Schmidt norm of the general fast-forward Hamiltonian (4) is given by

$$\|\hat{H}_{\text{FF}}(t)\|^2 = \sum_{\sigma} \left( \hbar \frac{df_{\sigma}(t)}{dt} - \frac{ds}{dt} \langle \sigma | \hat{H}(s) | \sigma \rangle \right)^2$$
$$+ \sum_{\substack{\sigma,\sigma' \\ (\sigma \neq \sigma')}} \left| i\hbar \left( 1 - e^{-i(f_{\sigma}(t) - f_{\sigma'}(t))} \right) \langle \sigma | \partial_t \sigma' \rangle - \frac{ds}{dt} \langle \sigma | \hat{H}(s) | \sigma' \rangle \right|^2$$

(12)

(see Appendix A for derivation). Generally, it is not an easy task to find optimal phase $f_{\sigma}(t)$ which minimizes this quantity, but we will show an example below.

We can find optimal phase $f_{\sigma}(t)$ when the measurement basis $\{|\sigma\rangle\}$ does not depend on time, i.e., when $\langle \sigma | \partial_t \sigma' \rangle = 0$. In this case, we can minimize Eq. (12) by setting the phase $f_{\sigma}(t)$ so that it satisfies

$$\hbar \frac{df_{\sigma}(t)}{dt} = \frac{ds}{dt} \langle \sigma | \hat{H}(s) | \sigma \rangle.$$

(13)

Then, Eq. (12) is given by

$$\|\hat{H}_{\text{FF}}(t)\|^2 = \left( \frac{ds}{dt} \right)^2 \sum_{\sigma} \left( \langle \sigma | \hat{H}^2(s) | \sigma \rangle - \langle \sigma | \hat{H}(s) | \sigma \rangle^2 \right),$$

(14)

which is upper bounded as

$$\|\hat{H}_{\text{FF}}(t)\|^2 \leq \left( \frac{ds}{dt} \right)^2 \sum_{\sigma} \langle \sigma | \hat{H}^2(s) | \sigma \rangle.$$

(15)

Notably, the right-hand side of this inequality is the squared Hilbert-Schmidt norm of the simplest fast-forward Hamiltonian (1), and thus general fast-forward scaling can achieve nontrivial speedup with the energy costs

$$\delta C(t) \leq \delta C_\$(t), \qquad C \leq C_\$. \tag{16}$$

The former inequality says that we can obtain $(ds/dt)$-times faster dynamics without requiring a $(ds/dt)$-times higher instantaneous energy cost, and the latter inequality says that we can even save the total energy in spite of speedup. We refer to such a time-saving and energy-saving process as energy-saving fast-forward scaling.

## 5 Examples

Now we show some examples of energy-saving fast-forward scaling. Hereafter, we set $\hbar = 1$.

### 5.1 Two-level system

As the simplest example, we first consider a two-level system

$$\hat{H}(t) = \omega(t)\hat{Z} + \Gamma(t)\hat{X}, \tag{17}$$

where $\omega(t)$ and $\Gamma(t)$ are time-dependent parameters, and the Pauli matrices are expressed as $\{\hat{X}, \hat{Y}, \hat{Z}\}$. The squared Hilbert-Schmidt norm of this Hamiltonian is given by

$$\|\hat{H}(t)\|^2 = 2(\omega^2(t) + \Gamma^2(t)), \tag{18}$$

which will be used to calculate the energy costs.

As a measurement basis, we consider the Pauli-Z basis $|\sigma\rangle = |\pm 1\rangle$ where $\hat{Z}|\pm 1\rangle = \pm|\pm 1\rangle$. Then, the squared Hilbert-Schmidt norm of the fast-forward Hamiltonian (12) is given by

$$\|\hat{H}_{\text{FF}}(t)\|^2 = \sum_{\sigma=\pm} \left( \frac{df_\sigma(t)}{dt} - \sigma \frac{ds}{dt}\omega(s) \right)^2 + 2\left( \frac{ds}{dt} \right)^2 \Gamma^2(s), \tag{19}$$

and it is optimized as

$$\|\hat{H}_{\text{FF}}(t)\|^2 = 2\left( \frac{ds}{dt} \right)^2 \Gamma^2(s), \tag{20}$$

when the phase $f_\pm(t)$ satisfies

$$\frac{df_\pm(t)}{dt} = \pm \frac{ds}{dt}\omega(s). \tag{21}$$

Note that the simplest fast-forward scaling corresponds to $f_\pm(t) = 0$. The optimized instantaneous energy cost is given by

$$\delta C(t) = \left| \frac{ds}{dt} \right| \sqrt{\frac{\Gamma^2(s)}{\omega^2(s) + \Gamma^2(s)}}, \tag{22}$$

which is strictly smaller than $\delta C_\$(t) = |ds/dt|$ when $\omega(s)$ is nonzero [$\omega(s) = 0$ gives the simplest fast-forward scaling except for global phase because it results in $f_\pm(t) = \text{const.}$]. Similarly, the total energy cost is also strictly smaller than $C_\$$ except for the case where $\omega(s)$ is always zero.

The optimal fast-forward Hamiltonian is given by

$$\hat{H}_{\text{FF}}(t) = \frac{ds}{dt}\Gamma(s)[\cos(f_+(t) - f_-(t))\hat{X} - \sin(f_+(t) - f_-(t))\hat{Y}], \tag{23}$$

where $f_\pm(t)$ is the solution of Eq. (21). That is, it is off-diagonal in the Pauli-Z basis, i.e., the present measurement basis.

## 5.2 Quantum annealing

Next, we consider application of energy-saving fast-forward scaling to quantum annealing. The quantum annealing Hamiltonian is given by

$$\hat{H}(t) = \lambda(t)\hat{H}_P + (1 - \lambda(t))\hat{V}, \qquad (24)$$

where $\lambda(t)$ is a time-dependent parameter satisfying $\lambda(0) = 0$ and $\lambda(T) = 1$, $\hat{H}_P$ is a problem Hamiltonian, and $\hat{V}$ is a driver Hamiltonian [26, 27]. In this paper, we consider the following problem and driver Hamiltonians

$$\hat{H}_P = -\frac{1}{2}\sum_{\substack{i,j=1 \\ (i \neq j)}}^{N} J_{ij}\hat{Z}_i\hat{Z}_j - \sum_{i=1}^{N} h_i\hat{Z}_i, \qquad \hat{V} = -\Gamma\sum_{i=1}^{N}\hat{X}_i, \qquad (25)$$

where $J_{ij} = J_{ji}$ is the strength of interaction, $h_i$ is the strength of a longitudinal field, $\Gamma$ is the strength of a transverse field, and $\{\hat{X}_i, \hat{Y}_i, \hat{Z}_i\}_{i=1}^{N}$ is the set of the Pauli matrices. The squared Hilbert-Schmidt norm of the quantum annealing Hamiltonian is given by

$$\|\hat{H}(t)\|^2 = 2^N \left[ \lambda^2(t)\left(\sum_{\substack{i,j=1 \\ (i<j)}}^{N} J_{ij}^2 + \sum_{i=1}^{N} h_i^2\right) + N(1-\lambda(t))^2\Gamma^2 \right], \qquad (26)$$

which will be used to calculate the energy costs.

As a measurement basis, we consider the computational basis $|\sigma\rangle = |\sigma_1, \sigma_2, \ldots, \sigma_N\rangle$ where $\hat{Z}_i|\sigma_1, \sigma_2, \ldots, \sigma_N\rangle = \sigma_i|\sigma_1, \sigma_2, \ldots, \sigma_N\rangle$ and $\sigma_i = \pm 1$ $(i = 1, 2, \ldots, N)$. Then, we can obtain the same measurement outcome with the original quantum annealing process within a shorter time

$$|\langle\sigma_1, \sigma_2, \ldots, \sigma_N|\Psi_{\text{FF}}(T_{\text{FF}})\rangle|^2 = |\langle\sigma_1, \sigma_2, \ldots, \sigma_N|\Psi(T)\rangle|^2. \qquad (27)$$

The squared Hilbert-Schmidt norm of the fast-forward Hamiltonian is given by

$$\|\hat{H}_{\text{FF}}(t)\|^2 = \sum_{\sigma}\left[\frac{df_{\sigma}(t)}{dt} + \frac{ds}{dt}\lambda(s)\left(\sum_{\substack{i,j=1 \\ (i<j)}}^{N} J_{ij}\sigma_i\sigma_j + \sum_{i=1}^{N} h_i\sigma_i\right)\right]^2 + N2^N\left(\frac{ds}{dt}\right)^2(1-\lambda(s))^2\Gamma^2. \qquad (28)$$

Full optimization of this quantity introduces $(\hat{Z}_i\hat{Z}_j)$-type terms in the unitary (6), and it results in many-body and non-local interactions in the fast-forward Hamiltonian (4) as the result of the Baker-Campbell-Hausdorff expansion. Its realization is experimantally hard, and thus we rather consider mitigating this quantity by setting

$$\frac{df_{\sigma}(t)}{dt} = -\frac{ds}{dt}\lambda(s)\sum_{i=1}^{N} h_i\sigma_i, \qquad (29)$$

which results in

$$\|\hat{H}_{\text{FF}}(t)\|^2 = 2^N\left(\frac{ds}{dt}\right)^2\left(\lambda^2(s)\sum_{\substack{i,j=1 \\ (i<j)}}^{N} J_{ij}^2 + N(1-\lambda(s))^2\Gamma^2\right). \qquad (30)$$

The partly-optimized instantaneous energy cost is given by

$$\delta C(t) = \left|\frac{ds}{dt}\right| \sqrt{\frac{\lambda^2(s)\sum_{\substack{i,j=1\\(i<j)}}^{N}J_{ij}^2 + N(1-\lambda(s))^2\Gamma^2}{\lambda^2(s)\left(\sum_{\substack{i,j=1\\(i<j)}}^{N}J_{ij}^2 + \sum_{i=1}^{N}h_i^2\right) + N(1-\lambda(s))^2\Gamma^2}}, \tag{31}$$

which is strictly smaller than $\delta C_\$(t) = |ds/dt|$ when $\{h_i\}$ is nonzero (the reason is similar to the previous example). Similarly, the total energy cost is also strictly smaller than $C_\$$ except for the case where $\{h_i\}$ is always zero.

The suboptimal fast-forward Hamiltonian is given by

$$\begin{aligned}
\hat{H}_{\text{FF}}(t) = &-\frac{ds}{dt}\lambda(s)\sum_{\substack{i,j=1\\(i<j)}}^{N}J_{ij}\hat{Z}_i\hat{Z}_j \\
&-\frac{ds}{dt}(1-\lambda(s))\Gamma\sum_{i=1}^{N}\left[\cos\left(2\int_0^s ds'\lambda(s')h_i\right)\hat{X}_i + \sin\left(2\int_0^s ds'\lambda(s')h_i\right)\hat{Y}_i\right],
\end{aligned} \tag{32}$$

where the phase of rotating fields $2\int_0^s ds'\lambda(s')h_i$ arises from the solution of Eq. (29). We notice that the longitudinal fields, which are diagonal in the computational basis, i.e., the present measurement basis, are replaced with rotating fields, which are off-diagonal in the computational basis.

## 5.3 Time-dependent measurement basis: lessons from a two-level system with the energy-eigenstate basis

Finally, we discuss the influence of a time-dependent measurement basis. As mentioned in Sec. 4, it is generally a hard task to find optimal phase $f_\sigma(t)$. Therefore, we try to find lessons from a simple example.

We consider the two-level system (17) and introduce the energy-eigenstate basis at the rescaled time as a time-dependent measurement basis, i.e., $|\sigma\rangle = |E_\pm(s)\rangle$ where $\hat{H}(s)|E_\pm(s)\rangle = E_\pm(s)|E_\pm(s)\rangle$ with $E_\pm(s) = \pm\sqrt{\omega^2(s)+\Gamma^2(s)}$. It corresponds with fast-forward scaling applied to nonadiabatic transitions [25]. The squared Hilbert-Schmidt norm of the fast-forward Hamiltonian is given by

$$\|\hat{H}_{\text{FF}}(t)\|^2 = \sum_{\sigma=\pm}\left(\frac{df_\sigma(t)}{dt} - \frac{ds}{dt}E_\sigma(s)\right)^2 + 4\left(\frac{d\theta(s)}{dt}\right)^2[1-\cos(f_+(t)-f_-(t))], \tag{33}$$

where

$$\begin{aligned}
\frac{d\theta(s)}{dt} &= \frac{ds}{dt}\langle E_+(s)|\partial_s E_-(s)\rangle \\
&= \frac{ds}{dt}\frac{\Gamma(s)\frac{d\omega(s)}{ds} - \omega(s)\frac{d\Gamma(s)}{ds}}{2(\omega^2(s)+\Gamma^2(s))}.
\end{aligned} \tag{34}$$

Notably, the quantity $d\theta(s)/dt$ is identical with a counterdiabatic field for the two-level system (17) (see, Refs. [12–14]). It means that the second term becomes large when the energy gap closes, or in other words, when the system tends to be nonadiabatic.

Now, we introduce concrete parameters. We consider magnetization reversal by assuming that the longitudinal field and the transverse field are given by

$$\omega(t) = \omega_0 - \frac{2\omega_0 t}{T}, \qquad \Gamma(t) = \Gamma_0, \tag{35}$$

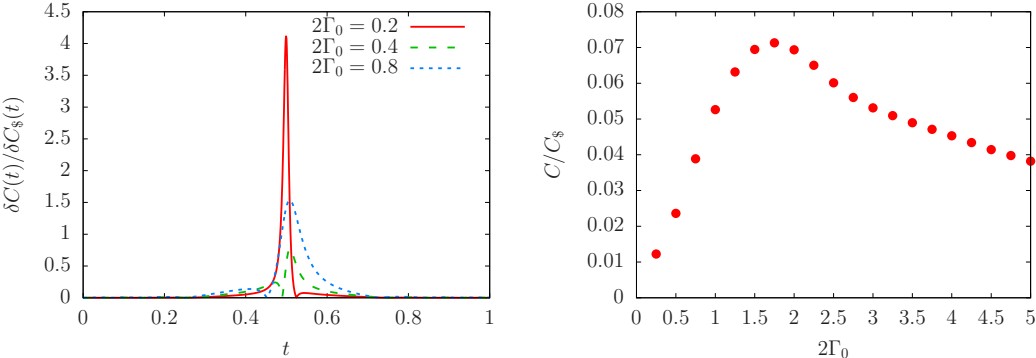

Figure 1: The instantaneous energy cost (left) and the total energy cost (right) against the standard counterparts. The instantaneous energy cost is plotted with respect to time for the energy gap $2\Gamma_0 = 0.2$ (red solid curve), 0.4 (green dashed curve), and 0.8 (blue dotted curve), and the total energy cost is plotted with respect to the size of the energy gap (red circles). The system parameters are given by $\omega_0 = 5$, $T = 10$, and $T_{\mathrm{FF}} = 1$.

where $\omega_0$ and $\Gamma_0$ are certain constants, and the rescaled time is given by linear rescaling

$$s(t) = \frac{T}{T_{\mathrm{FF}}} t \,. \tag{36}$$

The system crosses the energy gap at time $t = T_{\mathrm{FF}}/2$ and the size of the energy gap is $2\Gamma_0$. We are interested in behavior of the energy costs against the energy gap, and thus we fix other parameters as $\omega_0 = 5$, $T = 10$, and $T_{\mathrm{FF}} = 1$. That is, we consider 10-times faster control. Note that it is natural to adopt $\omega_0$ as a typical energyscale or $T$ as a typical timescale, but we adopt $T_{\mathrm{FF}} = 1$ as a typical timescale in numerical simulations for simplicity. Hereafter, all the temporal and energetic quantities are dimensionless in units of $T_{\mathrm{FF}} = 1$ and $T_{\mathrm{FF}}^{-1} = 1$ with $\hbar = 1$, respectively.

We optimize the first term only with Eq. (13), and then only the second term contributes to the energy costs. We depict the instantaneous energy cost and the total energy cost in Fig. 1. Here, the instantaneous energy cost is plotted for $2\Gamma_0 = 0.2$, 0.4, and 0.8, and the total energy cost is plotted for $2\Gamma_0 = 0.25, 0.5, \dots, 5$. We notice that general fast-forward scaling with phase (13) significantly suppress the instantaneous energy cost except for the vicinity of the energy gap. The spike of the instantaneous energy cost may seem irregular, but it consists of the monotonic envelope due to Eq. (34) and oscillating term $[1 - \cos(f_+(t) - f_-(t))]$ [see, Eq. (33)]. The latter oscillating term gives different shapes of the peaks. We also find that the total energy cost is significantly suppressed. The behavior of the total energy cost against the energy gap is not monotonic because it depends on the timing of oscillation $[1 - \cos(f_+(t) - f_-(t))]$ in the envelope due to Eq. (34).

We will additionally consider (sub)optimization of the second term to suppress the instantaneous energy cost in the vicinity of the energy gap. It can achieved when

$$f_+(T_{\mathrm{FF}}/2) - f_-(T_{\mathrm{FF}}/2) = 2\pi k \,, \tag{37}$$

with an integer $k$. To satisfy this condition, we modulate the phase (13) as

$$\frac{df_\sigma(t)}{dt} = \sigma(1 + \delta) \frac{T}{T_{\mathrm{FF}}} \sqrt{\left(\omega_0 - \frac{2\omega_0 t}{T_{\mathrm{FF}}}\right)^2 + \Gamma_0^2} \,, \tag{38}$$

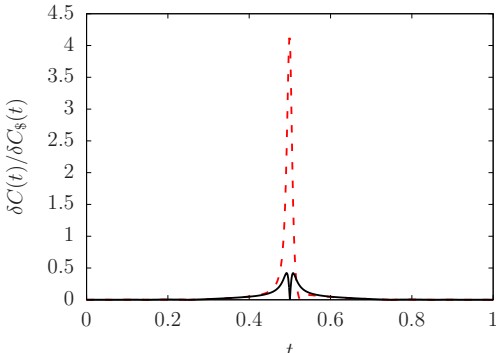

Figure 2: The instantaneous energy cost against the standard instantaneous energy cost. The horizontal axis is time. The black solid curve represents a scheme with modulation $\delta = 0.00326$ and the red dashed curve represents that without modulation. The system parameters are given by $\Gamma_0 = 0.1$, $\omega_0 = 5$, $T = 10$, and $T_{\mathrm{FF}} = 1$.

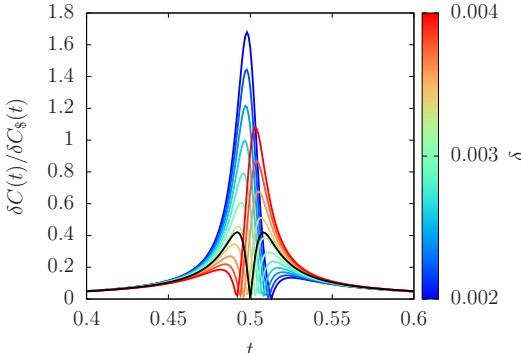

Figure 3: The $\delta$-dependence of the instantanous energy cost against the standard instantaneous energy cost. It is plotted from $\delta = 0.002$ to $\delta = 0.004$ by using gradient color curves from blue to red. The optimized one with $\delta = 0.00326$ is also plotted as the black curve. The horizontal axis is time. The system parameters are given by $\Gamma_0 = 0.1$, $\omega_0 = 5$, $T = 10$, and $T_{\mathrm{FF}} = 1$.

with small $\delta$. Note that $\delta$ should be small compared with the amplified eigenenergy $(ds/dt)E_\sigma(s)$ not to affect the optimality of the first term in Eq. (33). Then, the condition (37) says

$$\delta = \frac{1}{T} \frac{4\pi\omega_0 k}{\omega_0\sqrt{\omega_0^2 + \Gamma_0^2} + \Gamma_0^2 \log \frac{1}{\Gamma_0}(\omega_0 + \sqrt{\omega_0^2 + \Gamma_0^2})} - 1 \,, \tag{39}$$

and we notice that it gives $\delta \approx 0.00326$ when $k = 4$ in the present parameter setting with $\Gamma_0 = 0.1$. We depict the instantaneous energy cost of fast-forward scaling with this modulation in Fig. 2. We find that the peak in the instantaneous energy cost around the energy gap is suppressed because of the modulation of the oscillating term $[1 - \cos(f_+(t) - f_-(t))]$. In this way, we can also suppress the second terms in Eq. (33). We remark that similar suppression can also be found for other values of $\Gamma_0$.

Finally, we study the robustness of the above phase modulation. We plot $\delta$-dependence of the instantaneous energy cost against the standard one in Fig. 3. We find that the instantaneous energy cost gradually changes in the envelope, and thus we could realize energy-saving processes even if some errors take place in the parameter $\delta$.

# 6  Discussion

We found that the optimal fast-forward Hamiltonian of the two-level system (23) is off-diagonal in the Pauli-Z basis which was introduced as the time-independent measurement basis, and the suboptimal fast-forward Hamiltonian of the general Ising spin glass (32) requires suppression of some diagonal terms in the computational basis which was also introduced as the time-independent measurement basis. This is because the phase of fast-forward scaling $f_\sigma(t)$ can change the amplitude of diagonal terms in a measurement basis, while it does not affect the amplitude of off-diagonal terms when a measurement basis is independent of time [see, Eq. (A.2) in Appendix A]. That is, (sub)optimization of the energy costs can be realized by (sub)optimization of diagonal terms in a time-independent measurement basis. This property also explains why a measurement basis is important to save the energy costs despite the definitions of the energy costs do not depend on the choice of bases (the Hilbert-Schmidt norm is a basis-independent quantity).

Next, we discuss possible generalization of the lessons obtained in Sec. 5.3. We expect that similar results can be obtained when the time-dependent measurement basis is given by the energy-eigenstate basis at rescaled time. Here, we consider a general Hamiltonian

$$\hat{H}(t) = \sum_n E_n(t)|n(t)\rangle\langle n(t)|, \tag{40}$$

where $E_n(t)$ is the energy eigenvalue and $|n(t)\rangle$ is the corresponding energy eigenstate. Suppose that the measurement basis is given by the energy eigenstate, and then the squared Hilbert-Schmidt norm of the fast-forward Hamiltonian is given by

$$\|\hat{H}_{\text{FF}}(t)\|^2 = \sum_n \left(\frac{df_n(t)}{dt} - \frac{ds}{dt}E_n(s)\right)^2 \\ + 2\left(\frac{ds}{dt}\right)^2 \sum_{\substack{m,n \\ (m\neq n)}} \left|\frac{\langle m(s)|(\partial_s \hat{H}(s))|n(s)\rangle}{E_n(s) - E_m(s)}\right|^2 [1 - \cos(f_m(t) - f_n(t))]. \tag{41}$$

Here, the coefficient of the second term is the matrix element of the counterdiabatic Hamiltonian. Therefore, it results in the large instantaneous energy cost in the vicinity of the energy gap even if the first term is optimized, but we could mitigate it by slightly modifying the phase because of the oscillating term $[1 - \cos(f_m(t) - f_n(t))]$. Further study on this generalization and on other time-dependent measurement bases are left for the future work.

Experimental realization of energy-saving fast-forward scaling is of great interest. Recently, digital implementation of quantum annealing with the Pauli-Y terms was demonstrated by using superconducting and ion-trap quantum processors [28, 29], which is paid attention as the realization of digitized counterdiabatic driving [28–33]. All the examples discussed in Sec. 5 can experimentally be realized in a digital way. Because the Hilbert-Schmidt norm of a Hamiltonian, which is used in the energy costs, is related to the complexity of digital quantum simulation (see, e.g., Ref. [34]), energy-saving fast-forward scaling could be digitally simulated with smaller numbers of gate operations than the simplest fast-forward scaling if additional driving terms (the Pauli-Y operators in the examples discussed in the present paper) do not increase gate operations so much. We leave discussion on the performance of *digitized energy-saving fast-forward scaling* for the future work.

As discussed in Sec. 5.2, realization of fully-optimized energy-saving fast-forward scaling is generally a hard task. Indeed, even the phase factor consisting of the computational basis has $\hat{Z}_i\hat{Z}_j$-type interactions. For time-dependent bases or entangled bases, more complicated interactions would be included in the phase factors, which will result in many-body and nonlocal

fast-forward Hamiltonians. This expectation limits application of energy-saving fast-forward scaling. It is important for practical use to develop approximation theory for energy-saving fast-forward scaling, which approximately realize target dynamics with reduced energy costs.

## 7 Conclusion

In this paper, we introduced the instantaneous energy cost and the total energy cost of fast-forward scaling. The former quantity enables us to determine the maximum speed of fast-forward scaling with limited energy scale and the latter quantity enables us to evaluate energy efficiency of fast-forward scaling. As the result of the introduction of these energy costs, we found the existence of energy-saving fast-forward scaling for time-independent measurement bases, which enables us to realize speedup of target dynamics without requiring energy costs intuitively expected. We showed the examples of energy-saving fast-forward scaling by considering the two-level system and quantum annealing of the Ising spin glass. We also discussed the influence of the energy-eigenstate basis of the two-level system, which depends on time. We found that the instantaneous energy cost can significantly be suppressed except for the vicinity of the energy gap in the similar way to the case with the time-independent measurement basis. Moreover, we can mitigate the instantaneous energy cost around the energy gap by slightly modulating the phase. Notably, the total energy cost is quite low even without mitigation of the instantaneous energy cost. We also discussed interpretation, generalization, and experimental realization (limitation) of the present results. We believe that the present proposal and future followup enable us to realize energy-efficient quantum technologies.

## Acknowledgments

The author is grateful to Koji Azuma for useful comments.

**Funding information**  This work was supported by JST Moonshot R&D Grant Number JP-MJMS2061.

## A  Derivation of Eq. (12)

For a given complete orthonormal measurement basis $\{|\sigma\rangle\}$, the unitary operator (6) is rewritten as

$$\hat{U}_f(t) = e^{i\sum_\sigma f_\sigma(t)|\sigma\rangle\langle\sigma|} = \sum_\sigma e^{if_\sigma(t)}|\sigma\rangle\langle\sigma|, \tag{A.1}$$

and thus the general fast-forward Hamiltonian (4) is given by

$$\hat{H}_{\mathrm{FF}}(t) = \frac{ds}{dt}\sum_{\sigma,\sigma'} e^{i(f_\sigma(t)-f_{\sigma'}(t))}\langle\sigma|\hat{H}(s)|\sigma'\rangle|\sigma\rangle\langle\sigma'| \tag{A.2}$$

$$+\left(-\hbar\sum_\sigma \frac{df_\sigma(t)}{dt}|\sigma\rangle\langle\sigma| + i\hbar\sum_\sigma |\partial_t\sigma\rangle\langle\sigma| + i\hbar\sum_{\sigma,\sigma'} e^{i(f_\sigma(t)-f_{\sigma'}(t))}|\sigma\rangle\langle\partial_t\sigma|\sigma'\rangle\langle\sigma'|\right)$$

$$= \sum_\sigma \left(\frac{ds}{dt}\langle\sigma|\hat{H}(s)|\sigma\rangle - \hbar\frac{df_\sigma(t)}{dt}\right)|\sigma\rangle\langle\sigma|$$

$$+ \sum_{\substack{\sigma,\sigma'\\(\sigma\neq\sigma')}} \left[\frac{ds}{dt}e^{i(f_\sigma(t)-f_{\sigma'}(t))}\langle\sigma|\hat{H}(s)|\sigma'\rangle + i\hbar\left(1 - e^{i(f_\sigma(t)-f_{\sigma'}(t))}\right)\langle\sigma|\partial_t\sigma'\rangle\right]|\sigma\rangle\langle\sigma'|.$$

Then, by substituting this expression for

$$\begin{aligned}
\|\hat{H}_{\mathrm{FF}}(t)\|^2 &= \mathrm{Tr}\,\hat{H}_{\mathrm{FF}}^2(t) \\
&= \sum_{\sigma} \langle \sigma | \hat{H}_{\mathrm{FF}}^2(t) | \sigma \rangle \\
&= \sum_{\sigma,\sigma'} |\langle \sigma | \hat{H}_{\mathrm{FF}}(t) | \sigma' \rangle|^2 ,
\end{aligned} \tag{A.3}$$

we obtain Eq. (12).

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
