# Peer review of "Energy-saving fast-forward scaling"

_SciPost Physics, doi:SciPost Phys. Core 8, 019 (2025)_

## Round 2 · Referee Report · Anonymous (Referee 2) · 2024-10-3

Report

The revised manuscript has successfully addressed the majority of the comments and suggestions from the initial referee report, and I am satisfied with the changes made. For reference, below is a detailed evaluation of the points that were raised and whether they have been satisfactorily addressed. The overall quality of the work has improved, both in terms of clarity and depth of discussion. Therefore, I am pleased to recommend the manuscript for publication in SciPost Physics, pending any final editorial review.

$\textbf{Clarity in Sections IV and V (General, Comment 1):}$ The author has included additional equations and in-text references to improve the clarity and reproducibility of the arguments. The derivations are now more accessible.

$\textbf{Grammatical and Orthographic Corrections (General, Comment 2):}$ The manuscript has been thoroughly proofread, and the grammatical and typographical errors highlighted in the previous report have been addressed.

$\textbf{Upper Bounds on Speedup (Introduction, Comment 2): }$ The author has addressed the request to refer to limitations imposed by quantum speed limits, and relevant literature has been cited.

$\textbf{Magnification Factor and Slowing Down (Fast-forward scaling, Comment 1):}$ The author clarified the impact of the magnification factor and the possibility of applying the theory to cases where the dynamics is slowed down. This addition places the discussion on energy costs in a more general context.

$\textbf{Hilbert-Schmidt Norm Definition (Energy costs of fast-forward scaling, Comment 1):}$ A clear and concise definition of the Hilbert-Schmidt norm has been provided, ensuring there is no ambiguity in the energy cost metric used throughout the manuscript.

$\textbf{Measurement Basis and Energy Costs (Energy-saving fast-forward scaling Comment 1 and 2):}$ Appendix added to clarify derivation. The case where the measurement basis is the eigenbasis of the Hamiltonian has been explained further.

$\textbf{Examples (Comment 1 and 2):}$ The author has expanded the discussion on the choice of measurement basis and the impact of using a different optimized phase.

$\textbf{Characterization of the Small Parameter $\delta$ (Example, Comment 3):}$ The smallness of the parameter $\delta$ has been characterized clearly, along with its relation to the other energetic scales in the problem.

$\textbf{Figures and Discussion (Figures, Comment 1-3): }$ The figures have been updated with more comprehensive captions, and the spike in instantaneous energy cost is now better explained. The limiting behavior of the energy cost as the transverse field strength increases is discussed in greater detail, providing a clearer understanding of the results presented in the figures and confirming that the total energy cost is not monotonic, as it initially appeared. The figure showing the robustness of the phase modulation is notably improved and insightful.

$\textbf{Conclusion and Future Perspectives (Conclusions, Comment 1):}$ The limitations and potential extensions of the study have been discussed briefly in the conclusions. The inclusion of possible directions for future work, particularly regarding experimental realization, strengthens the final section of the manuscript.

Recommendation

Publish (easily meets expectations and criteria for this Journal; among top 50%)

---

## Round 2 · Referee Report · Anonymous (Referee 1) · 2024-10-7

Report

The authors have successfully addressed all points raised in the previous referee reports. What was a worthwhile contribution even before, has now also gained in clarity and depth. I recommend the work for publication in SciPost Physics.

Recommendation

Publish (easily meets expectations and criteria for this Journal; among top 50%)

---

## Round 2 · Referee Report · Anonymous (Referee 3) · 2025-1-4

Strengths

  1. Clear presentation of the concept of fast-forward scaling
  2. Results are clear and interesting

Weaknesses

Despite the relative simplicity of the calculations the presentation does not provide physical insights (relevant points which are either not discussed or only mentioned are: the role of the measurement basis on the energy cost? Limitations imposed by the minimal gap? Connection to quantum thermodynamics?)

Report

Due to the weaknesses I feel the paper is better placed in SciPoSt Physics Core.

Recommendation

Accept in alternative Journal (see Report)

---

## Round 2 · Author Response

I resubmit the manuscript entitled “energy-saving fast-forward scaling”. In accordance with the referee’s comments, the manuscript was revised. I am grateful to the referees for giving useful comments which help me to significantly improve the presentation of the results. My point-by-point replies can be found below. I believe that the present manuscript deserves to be published in SciPost Physics.

[Reply to the Report 1]

I am grateful to the referee 1 for carefully reading the previous manuscript and for giving useful comments. I revised the manuscript according to the comments. My replies to the comments are as follows.
Regarding the Weakness 1, energy-saving fast-forward scaling has two implications for experiments. Suppression of the instantaneous energy cost implies the possibility of further speedup of target dynamics with limited energy scale, and suppression of the total energy cost implies a possible improvement in energy efficiency of speedup. I mention these points in the revised manuscript.
Regarding the Weakness 2, analyses on energy costs of counterdiabatic driving were already reported in Ref.19-23 in the revised manuscript. Because fast-forward scaling can be applied to speedup of nonadiabatic dynamics as well as speedup of adiabatic time evolution, we can regard the present results as generalization (or extension) of the previous works. Note that invariant-based inverse engineering has the same mathematical structure with counterdiabatic driving, and thus energy costs of invariant-based inverse engineering should be similar to those of counterdiabatic driving.
Regarding the Weakness 3, I showed experimentally realizable application of energy-saving fast-forward scaling to quantum annealing of general Ising spin glass, which is specialization of adiabatic quantum computation. Here, the measurement basis was the computational basis because we are interested in measurement outcomes with the computational basis in quantum annealing. It should be noted that the choice of the measurement basis is not arbitrary; it must be determined by target dynamics and desired measurement outcomes. It is actually difficult if the measurement basis is the energy eigenstate basis because we have to know the energy eigenstates and entanglement in the energy eigenstates introduces complicated interacting terms in the unitary $U_f(t)$, which results in nonlocal and many-body interactions of the fast-forward Hamiltonian as the result of the Baker-Campbell-Hausdorff expansion.
Regarding the Weakness 4 and the annotated PDF file, I carefully revised the manuscript.
I believe that I addressed all the requested changes in the revised manuscript and it satisfies all the criteria for publication in SciPost Physics.

[Reply to the Report 2]

I am grateful to the referee 2 for carefully reading the previous manuscript and for giving useful comments. I revised the manuscript according to the comments. My replies to the comments are as follows.
Regarding the General comment (1), I added explanation for each result. Moreover, I added derivation of the main result in Appendix.
Regarding the General comment (2), I proofread the manuscript carefully, and revised it.
Regarding Introduction (1), I removed the concerned paragraph and added a new paragraph on quantum technologies to the revised manuscript.
Regarding Introduction (2), I removed the concerned sentence and mentioned the limitation due to quantum speed limits in the next paragraph.
Regarding Introduction (3), it is rephrased.
Regarding Fast-forward scaling (1), the magnification factor is given by $ds/dt$. In the previous manuscript, I wrote that the magnification factor is set as $ds/dt\ge1$, but it can be $ds/dt\ge0$. The present results hold even if slowing-down and pausing of target dynamics are included. Please see the revised manuscript.
Regarding Energy cost of fast-forward scaling (1), I added the definition of the Hilbert-Schmidt norm to the revised manuscript.
Regarding Energy-saving fast-forward scaling (1), I added derivation of the squared Hilbert-Schmidt norm of the fast-forward Hamiltonian in Appendix.
Regarding Energy-saving fast-forward scaling (2), I have tried to associate the present results with the stability of dynamics or control, but I could not find convincing interpretation. This is because the variance is given by using the measurement basis instead of the target dynamics. I can answer the second half of the question. The measurement basis can be the eigenbasis of a Hamiltonian as I showed in the third example. This is because $\langle\sigma|\partial_t\sigma^\prime\rangle$ is not zero for the eigenbasis of a Hamiltonian, or in other words, the squared Hilbert-Schmidt norm of the fast-forward Hamiltonian is not the variance of a Hamiltonian for the eigenbasis. If the eigenbasis is independent of time, there is no transition between the measurement basis, and thus $H_{\mathrm{FF}}(t)=0$ is enough for fast-forwarding and no contradiction exists.
Regarding Examples (1), I added explanations. Moreover, I added discussion on the (sub)optimal Hamiltonians in a new section 6 Discussion.
Regarding Examples (2), I added sentences which explain difficulty in experimental realization of the full-optimized fast-forward Hamiltonian. Moreover, I discussed experimental realization of the sub-optimal fast-forward Hamiltonian in the new section 6 Discussion. Since we considered the computational basis as the measurement basis, we can only modify the Pauli-Z terms (and global phase). Therefore, I do not think that there are other useful suboptimal phases. Moreover, although the measurement basis plays an important role to decrease the energy costs, it is determined by the target state and the desired measurement outcome, and thus we do not have the choice of the measurement basis in general. For example, in quantum annealing, the measurement basis must be the computational basis or energy-eigenstate basis, which are identical at the final time. We cannot generally exploit the choice of the measurement bases.
Regarding Examples (3), I addressed these points. Please see the revised manuscript.
Regarding Conclusions (1), I (mainly) addressed these points in the new section 6 Discussion. Please see the revised manuscript. As mentioned above, the present results can also be applied to the slowing-down case, $0\le ds/dt\le1$, and thus energy-saving “slow-down” scaling is also possible.
Regarding Figures (1), the behavior of the spike does not seem monotonic, but it consists of monotonic envelopes and oscillations. The spike for $2\Gamma_0=0.4$ is accidentally suppressed compared with that for $2\Gamma_0=0.8$ because of the difference in the oscillation terms. This behavior of the spike is in principle irregular because it is determined by the timing of the oscillation in the monotonic envelope. In addition to this point, the total energy cost is not monotonic. I plotted it for a wide range in the revised manuscript.
Regarding Figures (2), as mentioned above, the behavior of the total energy cost is not monotonic. I changed the plot range. Please find it in the revised manuscript. I am not interested in further large $\Gamma_0$ because there will be no nonadiabatic transition, i.e., transitions in the present measurement basis, and thus the optimal fast-forward Hamiltonian will be $\hat{H}_\mathrm{FF}(t)\approx0$, while the standard fast-forward Hamiltonian will be $\hat{H}_\mathrm{FF}(t)=(ds/dt)\hat{H}(s)\approx(ds/dt)\Gamma_0\hat{X}$.
Regarding Figures (3), I added the Figure 3 for the study of $\delta$-dependence of the instantaneous energy cost.
Regarding Figures (4), I addressed this point.
I believe that I addressed all the requested changes in the revised manuscript and it deserves to be published in SciPost Physics.

---

## Round 2 · List of Changes

A new section 6 Discussion is added and 7 Conclusion, which was originally 6 Summary, is also rewritten.
Derivation and explanation of equations are added.
Interpretation of figures is added.
Figure 3 is added to discuss robustness of optimization.
Typos and errors are corrected.
Vague sentences are removed.

---

## Editorial Decision

published